# Research on Anomaly Detection of Surveillance Video Based on Branch-Fusion Net and CSAM

**DOI:** 10.3390/s23031385

**Published:** 2023-01-26

**Authors:** Pengjv Zhang, Yuanyao Lu

**Affiliations:** School of Information Science and Technology, North China University of Technology, Beijing 100144, China

**Keywords:** video anomaly detection, C3D network, attention mechanisms, Bi-GRU

## Abstract

As the monitor probes are used more and more widely these days, the task of detecting abnormal behaviors in surveillance videos has gained widespread attention. The generalization ability and parameter overhead of the model affect how accurate the detection result is. To deal with the poor generalization ability and high parameter overhead of the model in existing anomaly detection methods, we propose a three-dimensional multi-branch convolutional fusion network, named “Branch-Fusion Net”. The network is designed with a multi-branch structure not only to significantly reduce parameter overhead but also to improve the generalization ability by understanding the input feature map from different perspectives. To ignore useless features during the model training, we propose a simple yet effective Channel Spatial Attention Module (CSAM), which sequentially focuses attention on key channels and spatial feature regions to suppress useless features and enhance important features. We combine the Branch-Fusion Net and the CSAM as a local feature extraction network and use the Bi-Directional Gated Recurrent Unit (Bi-GRU) to extract global feature information. The experiments are validated on a self-built Crimes-mini dataset, and the accuracy of anomaly detection in surveillance videos reaches 93.55% on the test set. The result shows that the model proposed in the paper significantly improves the accuracy of anomaly detection in surveillance videos with low parameter overhead.

## 1. Introduction

To maintain public safety, more and more surveillance probes are deployed at road intersections, shopping centers, subway entrances, and other public locations, resulting in a flood of surveillance videos [1,2]. Law enforcement officials usually spend a lot of time watching surveillance videos to find crimes, but still frequently miss important details. People urgently need an algorithm that automatically detects abnormal behaviors in surveillance video to lessen the workload of law enforcement officials while also dealing with abnormal criminal behaviors faster and more correctly [3,4,5].

Video anomaly detection methods can be divided into four categories: reconstruction-based method, prediction-based method, classification-based method, and regression-based method. The reconstruction-based method works by training normal video data to generate a distribution representation of normal data [6,7]. The prediction-based method assumes that a continuous normal video has a regular contextual connection and that future frames can be predicted by learning the dependencies [8]. However, there are no such dependencies between frames in abnormal videos [9]. The classification-based method treats the detection of anomalies as a classification problem [10]. After learning multiple distribution patterns of normal samples, samples that do not follow these distribution patterns are classified as abnormal. The above three categories of methods are suitable for frame-level detection of video, but not for large-scale detection [11].

Therefore, video anomaly detection is mostly performed by the regression-based method. The method’s idea is to use the anomaly score as an evaluation indicator. After setting an appropriate threshold, the video is considered abnormal if the anomaly score is above the threshold. The method is suitable for the large-scale video anomaly detection due to the simple structure. The regression-based method is employed in this paper’s multiple-instance learning (as shown in Figure 1). The C3D network is used in the feature extraction network for multiple-instance learning. Each convolutional layer of the C3D network convolves the input feature map directly in three dimensions, which leads to two problems [12]. Firstly, the large size and number of 3D convolutional kernels lead to high parameter overhead [13,14]. Secondly, the C3D network understands the input features from only one perspective because of the direct convolution of the feature map, which leads to poor generalization ability.

To address the problems, we propose a feature extraction network named “Branch-Fusion Net”. The multi-branch structure not only reduces parameter overhead but also allows the network to understand the input feature maps from different perspectives to improve the generalization ability.

The features extracted by networks usually contain features that are not relevant to the detection task [16]. To suppress useless features and enhance important features, we propose a simple yet effective Channel Spatial Attention Module (CSAM). The CSAM is an end-to-end generic module that can be seamlessly integrated into three-dimensional convolutional neural networks.

Since the convolutional neural networks are suitable for extracting local features [17], we combine the Branch-Fusion Net and the CSAM as the local feature extraction network and use the Bi-Directional Gated Recurrent Unit (Bi-GRU) for the global feature extraction.

The contributions of our work include:(1)We propose the Branch-Fusion Net, which not only greatly reduces parameter overhead, but also improves the generalization ability of feature extraction by understanding the input feature maps from multiple perspectives. The network achieves state-of-the-art performance for behavior recognition tasks on multiple benchmarks.(2)We propose a simple yet effective CSAM to ignore useless features during the model training. The CSAM focuses attention on key channels and spatial feature areas in turn. Adding the CSAM to the mainstream 3D convolutional neural networks can significantly improve the feature extraction effect.(3)We establish a surveillance video dataset called ‘Crimes-mini’ containing five categories of abnormal behaviors. The model we propose achieves the detection accuracy of 93.55% on the test set.

## 2. Related Work

### 2.1. Anomaly Detection Methods

There are four methods to deal with anomaly detection, including the reconstruction-based method, prediction-based method, classification-based method, and regression-based method. Table 1 shows the advantages and disadvantages of the four methods.

Reconstruction-based method. Hasan et al. propose two Auto-Encoder-based methods that can perform anomaly detection without supervision [18], but the abnormal samples are sometimes subject to reconstruction errors. To address the shortcoming, Gong et al. propose an improved Auto-Encoder [19], named the “Memory-augmented Auto-Encoder (MemAE)”. The method obtains the encoding from the encoder based on the input, and then uses it as a query to retrieve the most relevant memory item for reconstruction. To better remember normal samples, Park et al. propose a memory module that can be updated according to the scheme [20].

Prediction-based method. Medel et al. propose the Convolutional Long Short-Term Memory Network (Conv-LSTM) [21], which incorporates convolution operation into the Long Short-Term Memory Network (LSTM). Anomaly detection is achieved by reconstructing past frames and predicting future frames. Lu et al. combine the Variational Auto-Encoder (VAE) with the Conv-LSTM to propose the Convolutional Variational Recurrent Neural Network (Conv-VRNN) for generating future frames [22]. To address the possible blurring of future frames, Mathieu et al. [23] use alternating convolution and Rectified Linear Unit (RLU) to generate future frames and propose a method to fuse multiple feature learning strategies to generate clear future frames. Ye et al. propose a novel deep Predictive Coding Network (AnoPCN) [24], which implements anomaly detection by unifying reconstruction and prediction methods into a framework.

Classification-based method. Sabokrou et al. [25] propose a single classification-based video anomaly detection method inspired by the Generative Adversarial Network (GAN) for training models. Based on Sabokrou’s study, Wu et al. propose a deep single classification neural network that is capable of obtaining compact single-class classifiers considering only normal samples [26]. Xu et al. propose an adaptive intra-frame classification network [27]. The network extracts and classifies the input appearance and motion features into several sub-regions, and then classifies the sub-regions. If the test classification result of the sub-region is different from the true classification, the video is considered abnormal.

Regression-based method. Sultani et al. propose an anomaly detection method based on multiple-instance learning by scoring abnormal and normal video clips and then picking out the clips with the highest anomaly scores to train the model [28]. Since the hinge loss function used for multiple-instance learning training is not smooth, Kamoona et al. propose a loss function to make the model robust to output anomaly scores [29]. Zhu et al. output anomaly scores by feeding the calculated optical flow into a sequence-enhancement network [30].

Among the four methods, only the regression-based method is suitable for dealing with large-scale video data, which can be applied in practical scenarios. Therefore, we use multiple-instance learning of the regression-based method for anomaly detection. With the multiple-instance learning for anomaly detection, how to efficiently extract the features is crucial to the detection result.

### 2.2. Feature Extraction Network for Video

The networks for feature extraction in videos can be divided into a two-stream network architecture and a three-dimensional convolutional neural network architecture.

The three-dimensional convolutional neural network architecture was firstly proposed in [12]. The 3D convolution adds a temporal dimension to the 2D convolution to extract features in both temporal and spatial dimensions. The output of the 3D convolution is still a 3D feature map. Specifically, the 3D convolution moves in three dimensions: height, width, and depth, for multiple video frames. At each position, element-by-element multiplication and addition provide a value. Since the filter is sliding through 3D space, the output values are also arranged in 3D space. The C3D network is shown in Figure 2, and the size of each layer is shown in Table 2.

The three-dimensional convolutional neural network architecture can directly utilize three-dimensional convolution to extract spatiotemporal features, so the network structure is simple. However, the increase in the dimensionality of 3D convolution leads to a large number of parameters in the network.

The two-stream architecture (as shown in Figure 3) was firstly proposed in [31]. Spatial Stream Convnet and Temporal Stream Convnet are used to extract the spatial and temporal features of the video, respectively, and finally, the two networks are fused by late fusion. Specifically, the input of Spatial Stream Convnet is a single video frame, which is responsible for extracting the appearance features of the frames. Multiple optical stream frames are input into Temporal Stream Convnet. The addition of an optical stream makes it easier to capture motion information in the neural network, and directly provides the model with motion information between frames. However, the optical flow needs to be pre-extracted, and it leads to the spatiotemporal feature extraction being time-consuming.

## 3. The Network Architecture

To introduce our model, we show the architecture in Figure 4, and a summary of the size of each stage is shown in Table 3.

The Branch-Fusion Net consists of five stages: Conv1, Conv2, Conv3, Conv4, and Conv5. We combined the Branch-Fusion Net and the CSAM as the local feature extraction network and used the Bi-Directional Gated Recurrent Unit (Bi-GRU) for the global feature extraction.

### 3.1. Branch-Fusion Net

We noted that group convolution not only reduces the parameter overhead but also understands the input features from different perspectives after dividing different channels into groups, so we proposed the multi-branch structure based on group convolution. We firstly introduce how group convolution reduces the parameter overhead.

For the feature map of C × D × H × W, we can generate N × D × H × W by using N convolution kernels of C × d × h × w (assuming that the image size remains the same after pooling). Then, we assumed that the size of the input feature map was still C × D × H × W (Channel × Depth × Height × Width) and the number of output feature maps was N. If we divide it into 2 groups (as shown in Figure 5), the size of the input feature map is C/2 × D × H × W for each group, and the number of output feature maps of each group is N/2. The size of each convolution kernel is C/2 × d × h × w. The number of convolutional kernels in each group is N/2. The kernels only convolve with the input feature map of the same group. Therefore, the total number of convolutional kernels was N × C/2 × d × h × w. It can be seen that the total number of parameters was reduced to 1/2 of the original. When the number of groups is G, the number of parameters is reduced to 1/G.

Since each group in group convolution understands the input features independently and information does not circulate among different groups, this leads to the fact that each branch will understand the features from a different perspective.

Since group convolution groups input features from channel dimensions, which is not conducive to extracting global channel features, we improved group convolution by proposing the multi-branch structure, as shown in Figure 6. The first convolutional layer consisted of 128(4 × 32) convolutional kernels of 1 × 1 × 1. The number of convolution kernels was greatly reduced compared to the number of channels of input features. The size of the input feature map of the Branch-Fusion Net block was 256 × 16 × 64 × 64 (Channel × Depth × Height × Width), and the output feature map of each group was 4 × 16 × 64 × 64. We used 1 × 1 × 1 convolution for global feature preservation and completed the dimensionality reduction by making the total number of all grouped convolution kernels smaller than the number of channels in the input feature map, which greatly reduced the number of parameters of the network.

The second layer is the group convolution layer, which consisted of 128(4 × 32) convolution kernels of 3 × 3 × 3. The number of channels in the input feature map is the same as the number of convolution kernels. In this layer, we performed group convolution of the input feature maps with branches as groups. The feature map of 128 × 16 × 64 × 64 was divided into 32 groups from the channel dimension, and each group had a feature map of 4 × 16 × 64 × 64. Each group of input features was convolved by 3 × 3 × 3 to obtain an output feature map of 4 × 16 × 64 × 64. By group convolution, we further reduced the number of parameters of the network and improved the generalization ability of the network by understanding the input feature from multiple perspectives.

### 3.2. Channel-Spatial Attention Module

The CSAM includes the Channel Attention Module (CAM) and the Spatial Attention Module (SAM), as shown in Figure 7. The CAM is used to focus attention on the channels that have a greater impact on the final result at different times, and the SAM is used to focus attention on the regions of temporal and spatial features that are favorable for classification, which is complementary to the CAM. For the intermediate feature map of three-dimensional convolutional neural networks, the CSAM generates two attention maps along two independent dimensions, the channel and spatial, and then multiplies the attention by the input feature maps to perform adaptive feature refinement on the input feature maps.

#### 3.2.1. Channel Attention Module

In the convolutional neural network, each convolutional kernel has a different impact on the features. We used channel attention to focus our attention on the channel that has a great impact on the final result at different times. The CAM consists of two pooling layers, a multilayer perceptron, and an activation function. The structure is shown in Figure 8.

In Figure 8, ⊕ denotes the bitwise summation and 
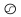
 denotes the sigmoid activation function. Firstly, the spatiotemporal information of the input features was aggregated using three-dimensional average pooling and three-dimensional maximum pooling to generate two different spatiotemporal context descriptors. Then, the descriptors were fed into a multilayer perceptron with shared weights to obtain two feature maps. Finally, the two feature maps were summed element-by-element and activated by the sigmoid function to obtain the final channel attention weights. The CAM is computed as:(1)$$M_{c}(F)=\sigma (MLP(AvgPool3d(F))+MLP(MaxPool3d(F)))$$
where *MLP* denotes the two-layer neural network and the sigmoid activation function. Since the input was a video sequence containing temporal information, the CAM was three-dimensional. The input was changed from (batchsize, channel, height, width) to (batchsize, channel, time_sequential, height, width).

#### 3.2.2. Spatial Attention Module

The SAM was designed to focus attention on regions of spatiotemporal features that are favorable for classification, and it complements the CAM. The SAM consists of two pooling layers, a convolutional layer, and an activation function. The structure is shown in Figure 9.

Firstly, the channel information of the input features was aggregated using average pooling and maximum pooling to generate two different channel context descriptors. Then, the two different channel context descriptors were stitched together, and the information was aggregated by the convolutional layer with the convolutional kernel of 7 × 7 × 7. Finally, the attention weights were obtained by activating the sigmoid function. The spatial attention is calculated as:(2)$$M_{s}(F)=\sigma (f^{7\times 7\times 7}([AvgPool(F);MaxPool(F)]))$$
where *f*^7×7×7^ denotes the convolution kernel of 7 × 7 × 7 and σ denotes the sigmoid activation function. Similar to the CAM, the average pooling, maximum pooling, and convolution in spatial attention were three-dimensional.

### 3.3. Bi-GRU

We first introduce the Gated Recurrent Unit (GRU), as shown in Figure 10b.

The calculation process in the GRU is summarized as follows:(3)$$z_{t}=\sigma (W_{z}•[h_{t-1},x_{t}])$$
(4)$$r_{t}=\sigma (W_{r}•[h_{t-1},x_{t}])$$
(5)$$\tilde{h_{t}}=\tanh(W•[r_{t}*h_{t-1},x_{t}])$$
(6)$$h_{t}=(1-z_{t})*h_{t-1}+z_{t}*\tilde{h_{t}}$$

The Gated Recurrent Unit (GRU) uses two gate functions to ensure that important features are not lost after long-term propagation. *z_t_* is the update gate responsible for controlling the amount of data that can be saved to the current moment in the forward parameter information, and *r_t_* is a reset gate to control how much of the past information to forget.

The idea of the Bi-Directional Gated Recurrent Unit (Bi-GRU) is to ensure the features obtained at time *t* have both the past and the future information. The network was divided into two independent GRUs, and the input sequences were input to the two GRUs in forward and reverse order for feature extraction. The two output vectors were stitched together to form the final feature representation. The structure of the Bi-GRU is shown in Figure 10a.

## 4. Experiment

### 4.1. Dataset

UCF-Crimes is a large dataset containing 128 h of surveillance video. It includes 1900 surveillance videos, divided into 14 categories of behavior. There exist some serious problems in the dataset, such as the low resolution and repetition of some video frames. In this paper, we selected five categories in UCF-Crimes: arson, burglary, explosion, road accident, and stealing, and established the Crimes-mini dataset after the preprocessing operations, including cropping, de-duplication, regrouping, and expansion. The training set, validation set, and test set were randomly selected in the ratio of 6:2:2. Some of the sample images in the dataset are shown in Figure 11. The number of videos of each anomaly in our dataset is shown in Table 4.

### 4.2. Evaluation Index

In this paper, we chose the accuracy rate as the important evaluation metric for our experiments. Since the overall performance of the model is also critical, we considered parameter overhead and F1 values in our analysis. The accuracy and F1 values are shown in Figure 12.

Accuracy is calculated as:(7)$$acc=\frac{TP+TN}{TP+TN+FN+TN}\times 100\%$$

*F*1 value is calculated as:(8)F1=2•Precise•RecallPrecise+Recall

Recall is calculated as:(9)Recall=TPTP+FN

Precise is calculated as:(10)$$\mathrm{Pr}ecise=\frac{TP}{TP+FP}$$

Parameter overhead is used to measure the size of the model. The lower the parameter overhead, the fewer parameters required to save the model. Parameter overhead of the 3D convolutional layer is calculated as:(11)$$params=C_{0}\times (k_{w}\times k_{h}\times k_{d}\times C_{i}+1)$$
where *C*_0_ denotes the number of output channels, *C_i_* denotes the number of input channels, *k_w_* denotes the convolutional kernel width, *k_h_* denotes the convolutional kernel height, *k_d_* denotes the length of the convolutional kernel in the time dimension, +1 denotes bias, brackets denote the number of parameters of a convolutional kernel, and *C*_0_ denotes that the layer has *C*_0_ convolutional kernels.

Parameter overhead of the fully connected layer is calculated as:(12)params=(I+1)×O=I×O+O
where *I* × *O* denotes the number of weights of *O* layers, and the number of bias is *O*.

### 4.3. Hyperparameter Setting

The setting of hyperparameters is critical, and it directly affects how well the model is trained. Therefore, we experimented with the setting of hyperparameters on the validation set.

#### 4.3.1. Number of Branches

We used the Branch-Fusion Net as the local feature extraction network. The generalization ability of the network directly affects our detection accuracy, and the number of branches determines the generalization ability. Therefore, we set the number of branches to 16, 24, 32, and 48, and analyzed the effects of different branch numbers on the generalization ability. The experimental result for different branch numbers is shown in Figure 13.

The experimental result shows that the detection accuracy was the highest when the number of branches was 32. When the number of branches was 16 or 24, the lack of generalization ability of the network led to low accuracy due to the small number of branches. When the number was 48, the accuracy was slightly lower than 32 branches, and the parameter overhead was the largest. Therefore, we chose 32 as the number of branches.

#### 4.3.2. Learning Rate Setting

The learning rate is an important hyperparameter when we train the model. For the gradient descent method, the choice of learning rate is critical. If it is too large, it will cause the model to fail to converge, and if it is too small, it will converge too slowly. We conducted experiments at different learning rates, and the experimental result is shown in Figure 14.

As can be seen from the figure, when the number of branches was 32, the F1 value was generally higher than in the other three cases, which is consistent with the results of our previous experiment. The accuracy rate and F1 value reached the maximum when the number of branches was 32 and the learning rate was 0.0005, which were 93.22% and 89.87%, respectively.

#### 4.3.3. Number of Layers of Bi-GRU

We used the Bi-GRU to accomplish the global feature extraction, so its performance is also critical to the overall model. We trained Bi-GRU on the test set for 60 rounds in different layers. The result is shown in Figure 15.

The experimental results show that global features extracted using the single-layer network are not enough, while the features extracted by the three layers are too abstract. Both will interfere with the subsequent training and make it less effective. When Bi-GRU used two layers, it could effectively complete the extraction of global complex information.

### 4.4. Structure Setting of CSAM

The CSAM includes two modules, the CAM and the SAM. To find the optimal connection for these two modules, we experimented with behavior recognition tasks with different connections using Branch-Fusion Net on the test set, and the experimental result is shown in Table 5.

According to the data in the table, adding either the CAM or the SAM to Branch-Fusion Net alone will improve the performance of the network, but the combination of the two is more effective than using individual modules. After connecting the two in series and parallel, we found that the “CAM + SAM” series connection was the most effective.

### 4.5. Comparison Experiments with Branch-Fusion Net

To evaluate the generalization capability of Branch-Fusion Net, we reproduced the mainstream feature extraction networks on the multiple benchmarks (UCF-101, HMDB51, and Kinetics) to conduct a comparison experiment for the behavior recognition task [32]. The Kinetics dataset is much larger than the UCF-101 dataset and the HMDB51 dataset, so it is more challenging. Therefore, we removed the networks (two-stream, C3D, and R3D) with poor feature extraction ability on the UCF-101 dataset and HMDB51 dataset, and then showed the TOP-1 accuracy and TOP-5 accuracy of the remaining networks on the Kinetics dataset. The experimental result is shown in Table 6 and Table 7.

The experimental result shows that our proposed Branch-Fusion Net can extract features better than the mainstream networks with two-Stream architecture (two-stream, TSN, TSM, TEA, TDN, and SlowFast) and three-dimensional convolutional architecture (C3D, R3D, R(2 + 1)D, S3D, two-stream I3D, X3D, NL I3D). We can see that the accuracy of our proposed Branch-Fusion Net was the highest on the small datasets of UCF-101 and HMDB51, reaching 97.32% (UCF-101) and 82.14% (HMDB51), which indicates that Branch-Fusion Net can fully understand the features on these two datasets. On the larger and more challenging Kinetics dataset, the Branch-Fusion Net also achieved state-of-the-art results, with 80.63% and 93.81% for TOP-1 and TOP-5, indicating that the Branch-Fusion Net can understand a large number of complex features.

### 4.6. Ablation Experiment of CSAM

The suppression of useless features and the highlighting of useful features will be reflected in the feature representation capability of the network with the CSAM added, and we chose the main 3D convolutional neural networks for the ablation experiment. The result is shown in Table 8.

The experimental result shows that the parameter overhead hardly increased after adding the CSAM to the networks, but there was a significant improvement in the accuracy rate, which indicates that the CSAM outperformed all the baselines without bells and whistles, demonstrating the general applicability of the CSAM across different architectures.

### 4.7. Comparison Experiment with Our Model

To evaluate the performance of our models (the Branch-Fusion Net with CSAM for the local feature extraction network and the Bi-GRU for the global feature extraction network), we conducted a comparison experiment with good feature extraction networks (P3D [43], R3D, and R (2 + 1)D) on the test set. The experimental result is shown in Figure 16.

The experimental results show that our proposed model achieved the highest accuracy with the lowest parameter overhead. The low parameter overhead indicates that the model requires low memory, and the 93.55% accuracy indicates that our proposed model can adequately extract the key features in abnormal videos.

## 5. Conclusions

We proposed a model based on multiple-instance learning, which included the local feature extraction network and the global feature extraction network. For the local feature extraction network, we proposed the Branch-Fusion Net to reduce parameter overhead and improve the generalization ability at the same time. To prevent useless features from interfering with the model training, we proposed the CSAM to suppress useless features and enhance important features. For the global feature extraction network, we used a two-layer Bi-GRU to complete the global feature extraction. To make the model more suitable for mobile devices, we will reduce the parameter overhead using model compression techniques in the future.

## Figures and Tables

**Figure 1 sensors-23-01385-f001:**
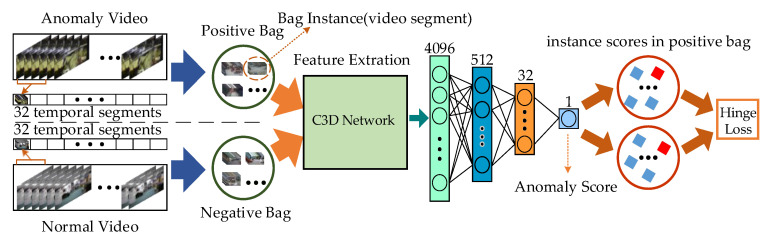
Multiple-instance learning. The surveillance video is divided into a fixed number of video segments, and the segments are placed in positive and negative bags. Each video segment is called an instance in the bags [15]. Then, the feature extraction network is used to extract features from the fixed number of video clips in the two bags. Next, the multilayer perceptron is used to calculate the anomaly scores of the features. Finally, the scores are used to determine whether there is abnormal behavior.

**Figure 2 sensors-23-01385-f002:**
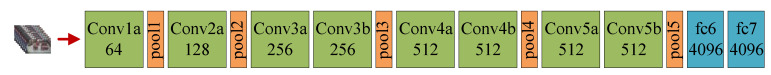
C3D network architecture.

**Figure 3 sensors-23-01385-f003:**
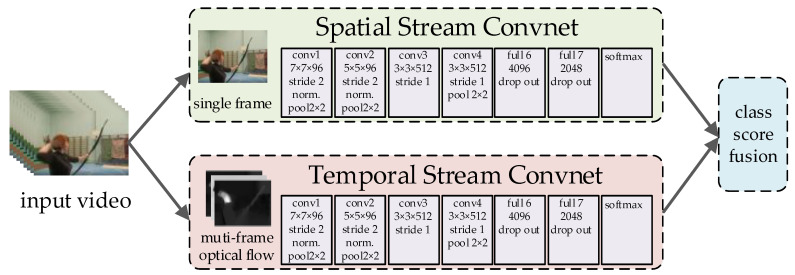
Two-stream network architecture.

**Figure 4 sensors-23-01385-f004:**
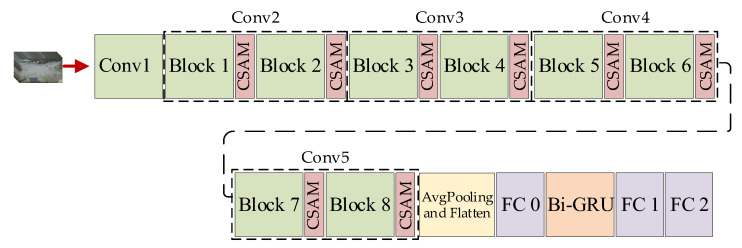
The architecture of our model.

**Figure 5 sensors-23-01385-f005:**
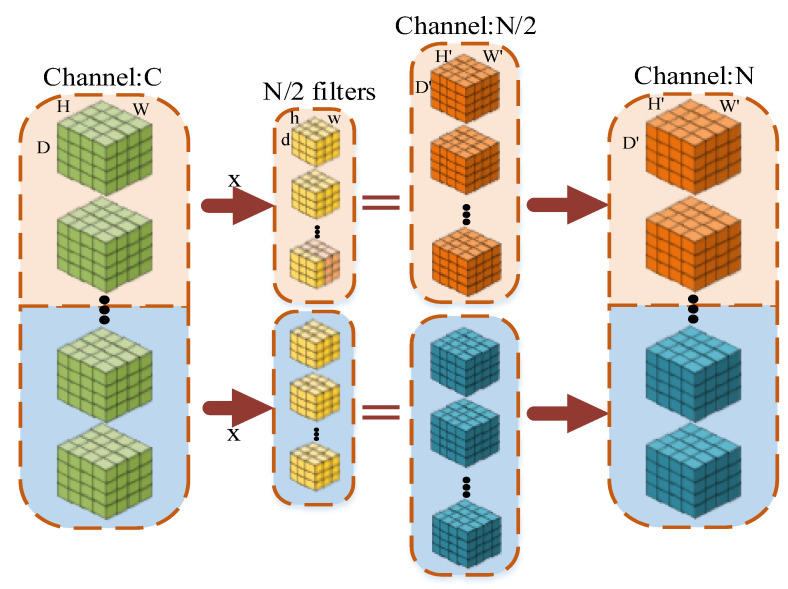
Group convolution diagram when the number of groups was 2.

**Figure 6 sensors-23-01385-f006:**
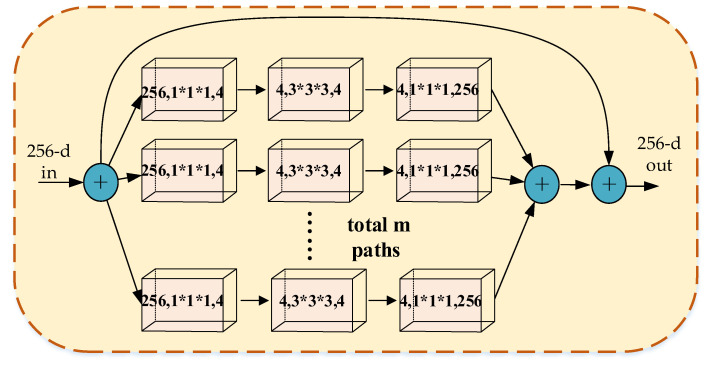
Block 2 of Branch-Fusion Net.

**Figure 7 sensors-23-01385-f007:**
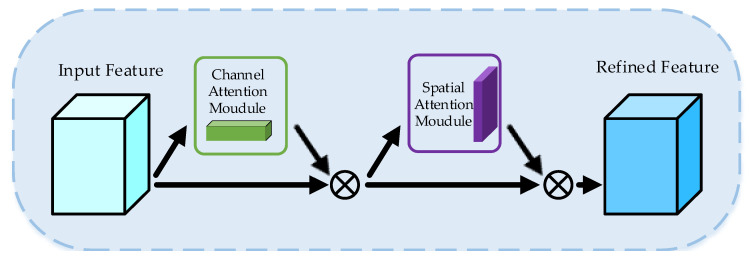
Channel Spatial Attention Module.

**Figure 8 sensors-23-01385-f008:**
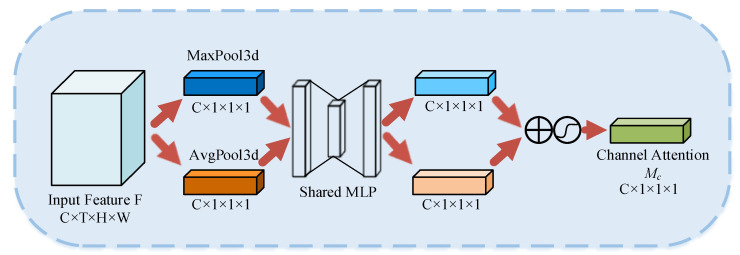
Channel Attention Module.

**Figure 9 sensors-23-01385-f009:**
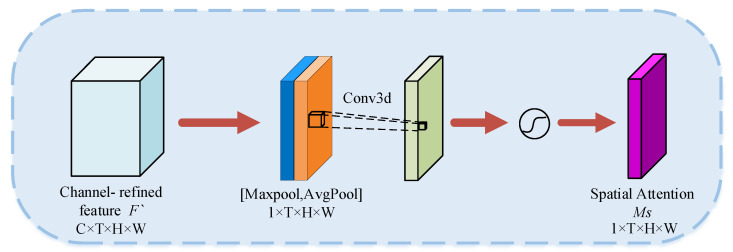
Spatial Attention Module.

**Figure 10 sensors-23-01385-f010:**
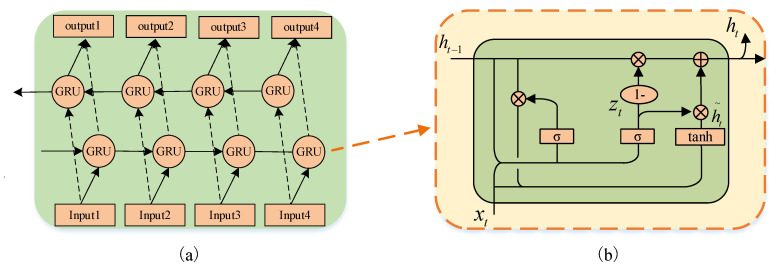
(**a**) The structure of Bi-GRU; (**b**) The structure of GRU.

**Figure 11 sensors-23-01385-f011:**
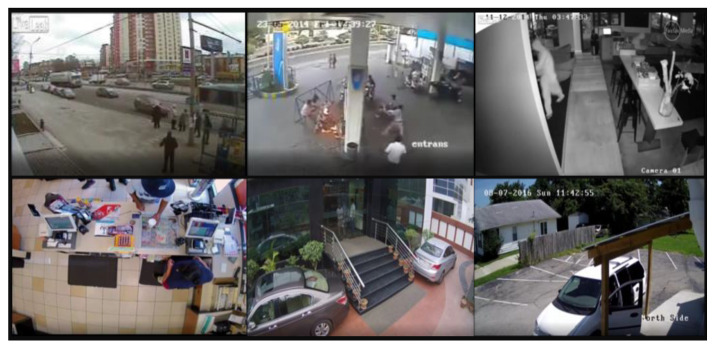
Sampling images with Crimes-mini dataset.

**Figure 12 sensors-23-01385-f012:**
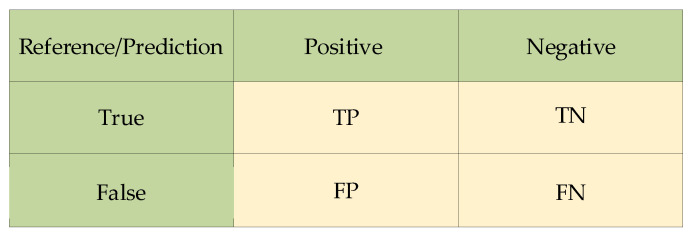
Confusion Matrix.

**Figure 13 sensors-23-01385-f013:**
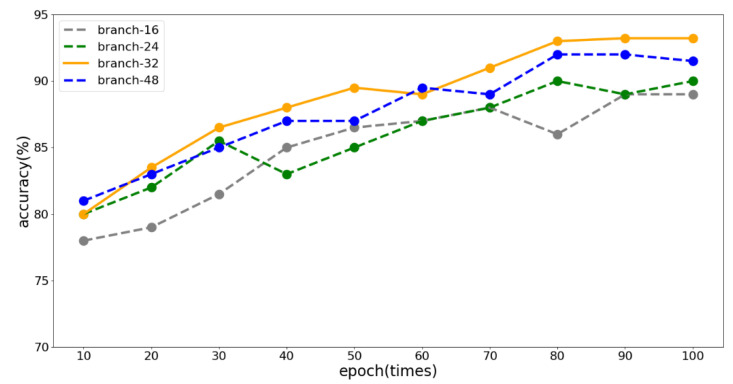
The effect of different numbers of branches on the accuracy rate.

**Figure 14 sensors-23-01385-f014:**
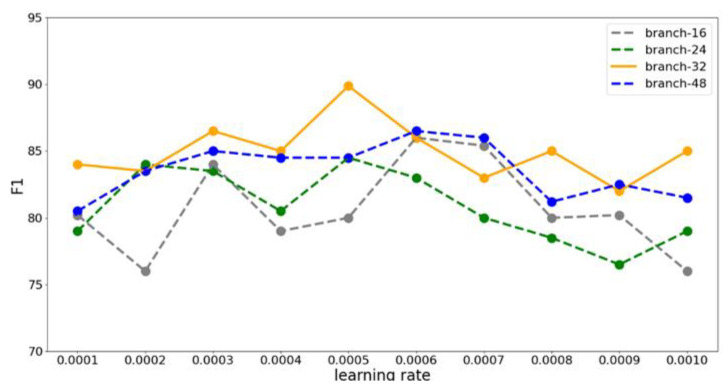
The effect of different learning rates on F1 values.

**Figure 15 sensors-23-01385-f015:**
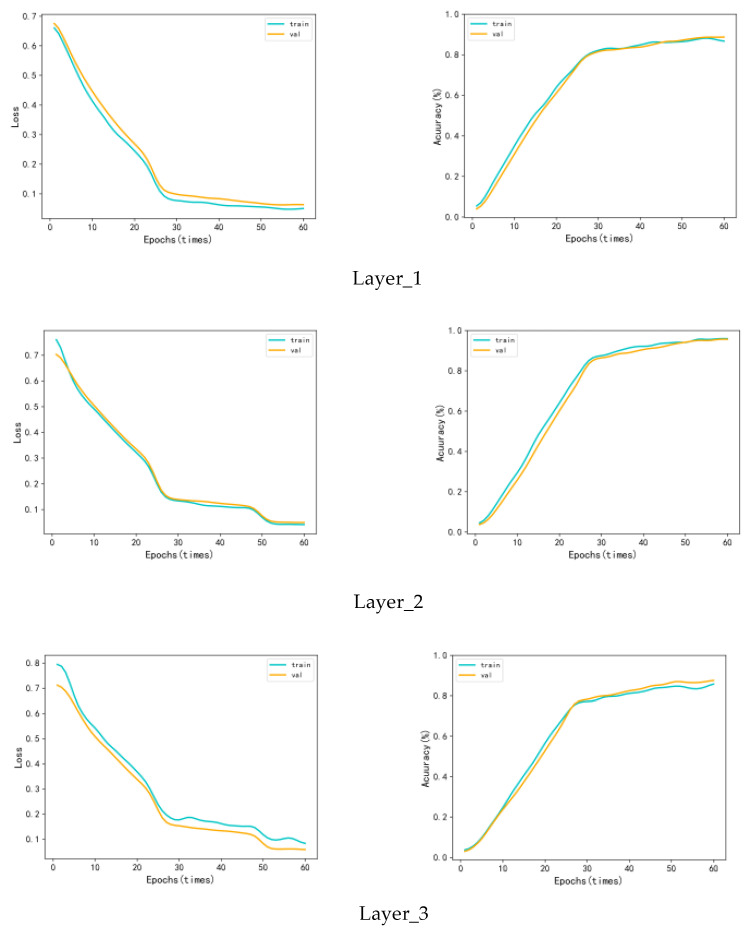
The training effect of Bi-GRU with different numbers of layers.

**Figure 16 sensors-23-01385-f016:**
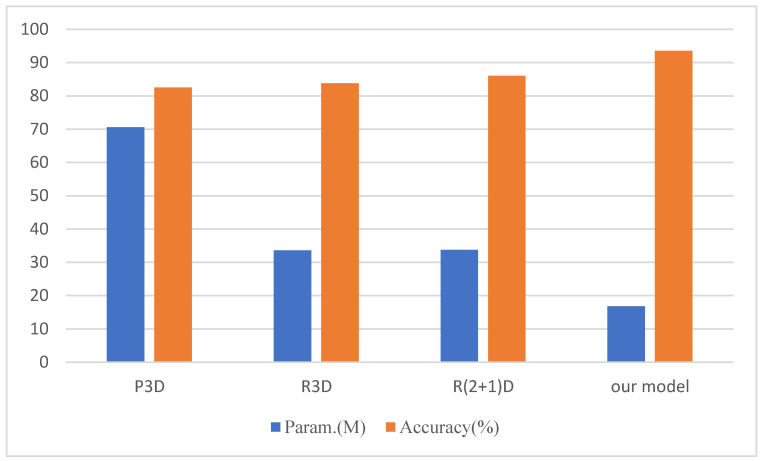
Comparison of parameter overhead and accuracy on the Crimes-mini dataset.

**Table 1 sensors-23-01385-t001:** Comparison and summary of different kind of methods.

Category	Judgment Basis	Advantages	Disadvantages
Reconstruction-based method	Small reconstruction errors for normal video frames and large reconstruction errors for abnormal video frames.	High accuracy in detecting location-related anomalies.	There are reconstruction errors for anomalous video frames.
Prediction-based method	Small prediction errors for normal video frames and large prediction errors for abnormal video frames.	High accuracy in detecting motion-related anomalies.	It ignores the fact that normal video frames can be unpredictable.
Classification-based method	The samples that do not follow the normal sample distribution are considered abnormal.	The distribution of normal samples can be well-learned.	If the normal sample distribution is complex, the classification model may fail.
Regression-based method	The samples with anomaly scores above the threshold are considered abnormal.	The model is simple and suitable for large-scale video anomaly detection.	The threshold for video anomalies is not easy to determine.

**Table 2 sensors-23-01385-t002:** The size of each layer in C3D network.

Layer Name	Size	Stride
Conv1a	3 × 3 × 3	1 × 1 × 1
pool1	1 × 2 × 2	1 × 2 × 2
Conv2a	3 × 3 × 3	1 × 1 × 1
Pool2	2 × 2 × 2	2 × 2 × 2
Conv3a	3 × 3 × 3	1 × 1 × 1
Conv3b	3 × 3 × 3	1 × 1 × 1
pool3	2 × 2 × 2	2 × 2 × 2
Conv4a	3 × 3 × 3	1 × 1 × 1
Conv4b	3 × 3 × 3	1 × 1 × 1
pool4	2 × 2 × 2	2 × 2 × 2
Conv5a	3 × 3 × 3	1 × 1 × 1
Conv5b	3 × 3 × 3	1 × 1 × 1
pool5	2 × 2 × 2	2 × 2 × 2

**Table 3 sensors-23-01385-t003:** The size of each stage in our model.

Stage	Output Size	The Size of the Stage
Conv1	64×8×28×28	3×3×3,64
Conv2	256×8×28×28	$\left[\begin{array}{c} 1\times 1\times 1,128\\ 3\times 3\times 3,128\\ 1\times 1\times 1,256 \end{array},C=32\right]\times 2$
Conv3	512×4×14×14	$\left[\begin{array}{c} 1\times 1\times 1,128\\ 3\times 3\times 3,128\\ 1\times 1\times 1,512 \end{array},C=32\right]\times 2$
Conv4	1024×2×7×7	$\left[\begin{array}{c} 1\times 1\times 1,256\\ 3\times 3\times 3,256\\ 1\times 1\times 1,1024 \end{array},C=32\right]\times 2$
Conv5	2048×1×4×4	$\left[\begin{array}{c} 1\times 1\times 1,512\\ 3\times 3\times 3,512\\ 1\times 1\times 1,2048 \end{array},C=32\right]\times 2$
AvgPooling and Flattening	1×2048	
FC0	1×300	2048×300
Bi-GRU	1×1×256	Hidden_size = 128 Num_layer = 2
FC1	1×128	256×128
FC2	1×4096	128×4096

**Table 4 sensors-23-01385-t004:** Number of videos of each anomaly in our dataset.

Anomaly	No. of Videos
Arson	100
Burglary	100
Explosion	100
Road accident	150
Stealing	120
Normal events	320

**Table 5 sensors-23-01385-t005:** Combining methods of channel and spatial attention.

Description	Accuracy (%)
Model (Branch-Fusion Net)	86.29
model + CAM	89.56
model + SAM	90.08
model + CAM and SAM in parallel	93.57
model + SAM + CAM	92.83
model + CAM + SAM	93.43

**Table 6 sensors-23-01385-t006:** Accuracy (%) comparison on UCF-101 and HMDB datasets.

Architecture	UCF-101	HMDB51
Two-Stream [31]	87.84	58.86
TSN [33]	93.90	70.88
TSM [34]	95.47	73.95
TEA [35]	96.63	73.12
TDN [36]	95.39	76.26
SlowFast [37]	96.20	78.04
C3D [12]	85.08	56.00
R3D [38]	85.22	53.80
R(2 + 1)D [39]	95.55	73.85
S3D [40]	96.62	75.33
X3D [41]	96.71	81.67
Two-Stream I3D [32]	97.29	80.71
NL I3D [42]	96.87	80.16
Branch-Fusion Net	97.32	82.14

**Table 7 sensors-23-01385-t007:** Accuracy (%) comparison on Kinetics dataset.

Description	TOP-1	TOP-5
TSN [33]	68.93	87.84
TSM [34]	74.39	90.89
TEA [35]	76.20	92.27
TDN [36]	76.85	93.16
S3D [40]	74.69	93.24
X3D [41]	79.06	93.76
R(2 + 1)D [39]	74.28	91.49
SlowFast [37]	77.11	92.37
Two-Stream I3D [32]	72.02	89.89
NL I3D [42]	76.38	92.55
Branch-Fusion Net	80.03	93.81

**Table 8 sensors-23-01385-t008:** Accuracy comparison on the Kinetics dataset.

Architecture	Params/M	Accuracy (%)
C3D	58.378	72.93
C3D + CSAM	58.454	77.71
R3D	33.642	80.29
R3D + CSAM	33.731	83.90
R(2 + 1)D	33.641	83.08
R(2 + 1)D + CSAM	33.730	88.44
Branch-Fusion Net	14.856	89.63
Branch-Fusion Net + CSAM	16.190	93.55

## Data Availability

Not applicable.

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
