# Peer review of "Research on Anomaly Detection of Surveillance Video Based on Branch-Fusion Net and CSAM"

_sensors, 2023, doi:10.3390/s23031385_

Round 1

Reviewer 1 Report

This paper proposes a model based on multiple instance learning, including local and global feature extraction networks. The paper combines Branch-Fusion Net and Channel-Spatial Attention Module (CSAM) as a local feature extraction network and applies Bi-Directional Gated Recurrent Unit (Bi-GRU) to extract the global feature information. The experiments are validated on the UCF-Crimes dataset. The result shows that the proposed model improves the accuracy of anomaly detection in surveillance videos with low parameter overhead. 

Major comments are as follows. 

1. Since the overall paper is poorly written, it is hard to understand the contributions of the paper. 

- It would be better to present the overall system diagram and architecture of Figures 1 and 2 in Section 3 rather than Section 1.

- Please reorganize and rewrite the whole parts of Section 3.

- Please improve the English grammar and check the spelling. 

2. The contributions of the paper are not clear. 

- The authors of the paper propose the Channel Spatial Attention Module (CSAM) to extract the local features. However, a similar idea has already been presented in [1]. 

- Section 3.2 Branch-Fusion Net includes many similar sentences and ideas from [2]. Please clarify the original contributions of the proposed scheme and rewrite the entire section. 

3. Please describe more details of the comparative studies in Section 4. 

 - While there are many comparative studies in Tables 3 and 4, the models are not well described. 

[1] Tao, W.; Xie, Z.; Zhang, Y.; Li, J.; Xuan, F.; Huang, J.; Li, X.; Su, W.; Yin, D. Corn Residue Covered Area Mapping with a Deep Learning Method Using Chinese GF-1 B/D High Resolution Remote Sensing Images. Remote Sens. 2021, 13, 2903. https://doi.org/10.3390/rs13152903

[2] Jinyuan Ni, Xinyue Zhang, Jianxun Zhang, "Multiscale Feature Fusion Attention Lightweight Facial Expression Recognition", International Journal of Aerospace Engineering, vol. 2022, Article ID 6523234, 15 pages, 2022. https://doi.org/10.1155/2022/6523234

Reviewer 2 Report

The paper proposes an interesting approach for anomaly detection on surveillance videos. It is, however, hard to judge the quality of the results, since the presentation needs to be heavily improved.

In particular, my concerns are:

- The model architecture is partly presented in Sec. 1 and partly in Sec. 3, these need to be joined into one Section. The overall presentation also needs to be improved, including the English language, since it is not always clear the goal of the model choices. For instance, Figure 1a is totally unclear, it becomes only slightly clearer after reading Sec. 3, but it still needs a better explanation.

- Sec. 2 is very chaotic, it needs complete rewriting, together with a table to summarize strengths and weaknesses of the presented models. Moreover, it should introduce the models (with abbreviations) that will then be used for comparison in the Results section.

- Regarding the branch structure, what is it that is divided into groups, is it the channels dimension? This is never clearly specified. And if this is the case, if the channel dimension is 3 (RGB cameras), how can there be up to 48 branches? This whole paragraph needs a much better explanation.

- The dataset description needs a much better explanation. How many examples of each anomaly are present? How are the clips divided into the bags structure defined in Fig. 2? How many data points are present in the end? How are these divided in training, validation and test?

- In between the lines, one assumes that the model validation in Sec. 4.3 - 4.4 is done on the validation set, however this is never specified (and the set never defined).

- All the models that are compared in Sec. 4.5 - 4.7 need to be properly referenced in the tables.

- In Sec. 4.8 it is completely unclear what the authors want to show. I would ask to either remove the section, or greatly expand it.

- Overall, the English language needs to be greatly improved. In particular the use of singulars and plurals and the correct use of the definite and indefinite articles.

Round 2

Reviewer 1 Report

The authors address the major comments of the reviewer. 

Reviewer 2 Report

The manuscript has significantly improved in the current version. There are still however some issues that need to be resolved:

1. There are still many detail missing on how the C3d network is combined with the Branch Fusion. Interpreting the text, I assume that the branches are not created for every convolutional layer, but only from the third layer on (with 128 channels). This is however never specified. There needs to be a detailed presentation, hopefully also with a sketch, that shows every details of the architecture, so that the results can be reproduced.

2. Additionally, it can still be not completely clear how all modules from Figs. 5-8 are integrated with each other and in the overall network. Also in this case, there needs to be a sketch that summarizes the overall architecture, and for the details it is enough to refer to the specific figures 5-8.

3. The architectures that the authors have chosen for comparison in Tabs 5-6 need to be properly defined in the Related work. Additionally, it has to be commented what are the pros and cons, at least with the couple of architectures that give a similar performance with respect to the author's model.

4. The improvement with respect to some models is minimal, if any, for instance with X3D[40]. This last model is also not present in Table 5, and the authors have to justify why they don't test it in this case, and if there is no real reason they should include it. Moreover, the authors have to clarify whether the results they show have been taken from the original papers, or reproduced by the authors on their dataset (which is the only correct thing to do in this case).
